# Methodology to Solve the Multi-Objective Optimization of Acrylic Acid Production Using Neural Networks as Meta-Models

Geraldine Cáceres Sepulveda [1], Silvia Ochoa [2] and Jules Thibault [1,*] 

[1]  Department of Chemical and Biological Engineering, University of Ottawa, Ottawa, ON K1N 6N5, Canada; gcace075@uottawa.ca

[2]  SIDCOP Research Group-Departamento de Ingeniería Química, Universidad de Antioquia, Medellín 050010, Colombia; silvia.ochoa@udea.edu.co

*  Correspondence: Jules.Thibault@uottawa.ca; Tel.: +1-613-562-8000

**Abstract:** It is paramount to optimize the performance of a chemical process in order to maximize its yield and productivity and to minimize the production cost and the environmental impact. The various objectives in optimization are often in conflict, and one must determine the best compromise solution usually using a representative model of the process. However, solving first-principle models can be a computationally intensive problem, thus making model-based multi-objective optimization (MOO) a time-consuming task. In this work, a methodology to perform the multi-objective optimization for a two-reactor system for the production of acrylic acid, using artificial neural networks (ANNs) as meta-models, is proposed in an effort to reduce the computational time required to circumscribe the Pareto domain. The performance of the meta-model confirmed good agreement between the experimental data and the model-predicted values of the existent relationships between the eight decision variables and the nine performance criteria of the process. Once the meta-model was built, the Pareto domain was circumscribed based on a genetic algorithm (GA) and ranked with the net flow method (NFM). Using the ANN surrogate model, the optimization time decreased by a factor of 15.5.

**Keywords:** multi-objective optimization; acrylic acid production; Pareto domain; artificial neural networks; surrogate model

## 1. Introduction

Given the highly competitive market in the chemical industries and the very high capital and operating costs of chemical manufacturing plants, it is paramount to constantly determine the optimal operating conditions, while considering economic, environmental, and societal constraints. However, formulating and solving an optimization problem in chemical engineering is not a straightforward task, because engineers will invariably have to deal with numerous conflicting/competing objectives. Therefore, there is a need to use optimization strategies that will provide the decision-maker with different alternative solutions that accurately reflect the existing underlying relationships between the process variables, which can be achieved via multi-objective optimization (MOO).

One of the most common approaches used for solving MOO problems is to convert them into a single-objective optimization problem, where an aggregated single objective function is built by taking the weighted sum of the different objectives [1]. However, this approach presents important drawbacks, including: (i) the optimal solution is a single point inside the feasible region; and (ii) this solution is highly sensitive to the selection of the weights assigned to each objective. In some cases, it may not even be possible to aggregate these competing objectives into a single objective, since some of the objectives are of a qualitative nature and it may be difficult to numerically quantifying them. Solving

a MOO problem explicitly can be very expensive in terms of the required computational load given the large number of model evaluations that are required to circumscribe the Pareto domain. Usually, one can develop a phenomenological model of the process to be used with the optimization algorithm, which is usually a very complex and time-consuming endeavor, to generate the Pareto-optimal front, which contains all the non-dominated solutions to the overall problem. Solutions are non-dominated when the improvement of any one of the objectives leads to the deterioration of at least one other objective [2,3]. One can also resort to a state-of-the-art process simulator to simulate the entire or part of the process while facing the challenge of interfacing the simulation software with the optimization algorithm. In addition, sometimes the computation time to perform one simulation run can be relatively long, such that the time to circumscribe the Pareto can take a significant amount of time for each optimization scenario. To reduce the simulation time and expedite the determination of the Pareto domain for a given optimization scenario, some authors have successfully used meta-models, also known as surrogate models, to represent the underlying relationships existing between the input or decision process variables and the objectives [4–11].

Artificial neural networks (ANNs) are often used as meta-models due their high plasticity to encapsulate underlying relationships within the process data. In training ANN models, the connection weights between two layers of neurons are adjusted as to minimize the sum of squares of the errors between the experimental and predicted outputs [12]. The main advantage of using ANNs as surrogate models is their capability of effectively model very complex nonlinear behavior, even if it includes a large number of variables. When trained, an ANN can be used to perform a large number of simulations and to determine the Pareto domain very rapidly. In this paper, a methodology for solving multi-objective optimization problems using artificial neural networks as meta-models is proposed and applied to a chemical engineering problem, namely the production of acrylic acid.

## 2. Description of the Acrylic Acid Production

The comprehensive optimization study of a process normally requires having an accurate and representative model. In this investigation, a first-principle based model was developed for the reactor section of an acrylic acid production plant, which was then used for generating the data in order to train and validate an ANN for solving the MOO. The first-principle based model was simulated in FORTRAN and validated successfully by comparing it with simulation results obtained from ASPEN and Honeywell UniSim. All the main considerations that were accounted for are described in the following sections.

### 2.1. Reactor Model

Acrylic acid (AA) plays an important role in the production of polymeric products. The worldwide production of acrylic acid reached ~3.9 million metric tons in 2009 [13]. This important chemical is used in the manufacture of superabsorbent polymers (SAPs) which are involved in a variety of applications [14]. A very large proportion of AA is converted to a wide range of esters that are applied in surface coatings, textiles, adhesives, paper treatment, polishes, plastics, and many others [13,15,16]. In a smaller proportion, it is also used for the production of detergents and flocculants [13].

Nowadays, the commercial production of AA comes from the petrochemical industry [15]. Currently, the preferred production route is by the partial vapor phase catalytic oxidation of propylene in a two-step process, as shown in Figure 1. In this process, propylene is first oxidized to acrolein, by supplying a mixture of propylene, air, and steam to the first reactor. Acrolein, which is an intermediate product, is subsequently oxidized to AA in the second reactor [16].

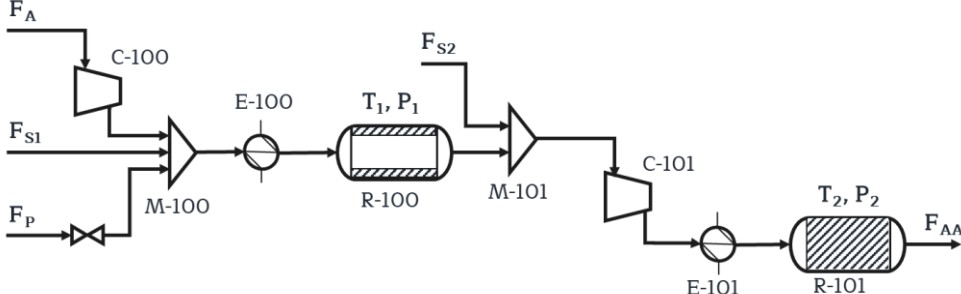

**Figure 1.** Process flow diagram of the two-reactor section of the production of acrylic acid.

It is important to note that the desired and undesired reactions are highly exothermic and highly temperature-dependent, such that steam, which is fed to the first and second reactors, acts as a thermal sink to moderate the rise in temperature. Furthermore, this process relies on two compressors to bring the reactor feed to the desired operating pressure and, as a result, higher air and water vapor flowrates will significantly increase the compression work. Another important consideration is the flammability of propylene in the first reactor, which will be addressed in the following sections.

### 2.2. Propylene Oxidation

The propylene vapor phase oxidation occurring in the first reactor is performed in a catalytic wall reactor (CWR), where the catalyst is coated on the inner surface of the reactor and it guarantees isothermal conditions through a temperature control loop for a range of 330–430 °C [17]. The oxide catalyst annular section of the CWR is constituted of bismuth and molybdenum containing montmorillonite.

The reaction scheme considered in the model formulation for this reactor is depicted in Figure 2, in which a total of 10 reactions involving propylene, oxygen, acrolein, acrylic acid, acetaldehyde, acetic acid, formaldehyde, carbon dioxide, and carbon monoxide were considered [18]. Given the very low concentrations of formaldehyde observed in the reactor outlet mixture following the series of experiments performed by Redlingshofer et al. [17], it was assumed to be negligible in this study, and was not considered in the model. As was previously mentioned, steam is present along with propylene and oxygen in the input stream. Steam plays an important role, by increasing the selectivity towards acrolein by suppressing the formation of carbon oxides at low temperatures and contributing to the catalyst re-oxidation. In addition, steam has a dilution effect that contributes not only as a thermal sink but also to ensure the reactor is operating below the flammability limits [18,19] (more information is provided in Section 2.4).

At temperatures higher than 360 °C, oxygen has a weak influence on the formation of acrolein and the catalytic reduction by propylene will be the rate-determining step. On the other hand, at temperatures lower than 360 °C, if the oxygen concentration increases, the formation of acrolein is accelerated. As a result, the expression for the reaction rate from propylene to acrolein will change depending on whether the temperature is below or above 360 °C [18].

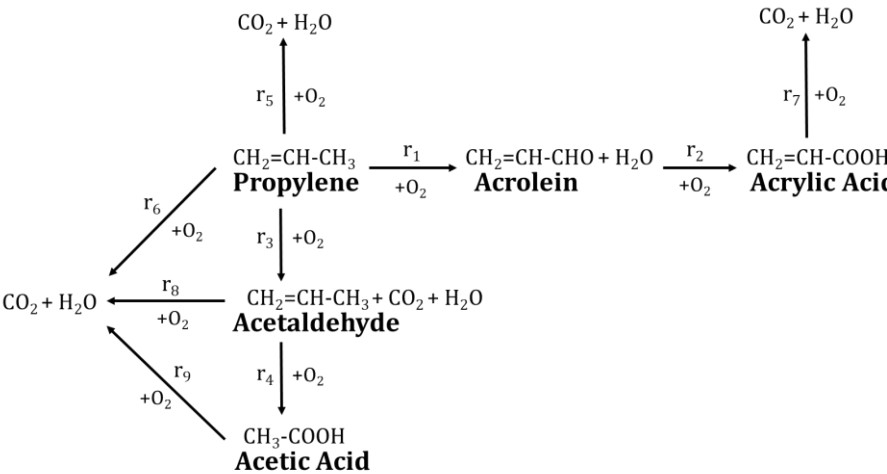

**Figure 2.** Reaction scheme for the propylene oxidation reactor (adapted from [18]).

### 2.3. Acrolein Oxidation

The acrolein vapor phase oxidation kinetics for the second reactor were determined using a catalyst that contains oxides of antimony, nickel, and molybdenum [20]. In this case, carbon dioxide was the only major by-product in the reaction scheme. Thus, only the formation of acrylic acid and carbon dioxide were used to determine suitable rate expressions [20].

$$C_3H_4O + \frac{1}{2}O_2 \overset{r_{10}}{\rightarrow} C_3H_4O_2 \tag{R1}$$

$$C_3H_4O + \frac{7}{2}O_2 \overset{r_{11}}{\rightarrow} 3CO_2 + 2H_2O \tag{R2}$$

The experimental study to obtain the kinetics for the acrolein oxidation was carried over a temperature range of 285 °C–315 °C [20]. The vapor phase acrolein oxidation reactor was assumed to take place in a packed bed reactor (PBR), and as the catalyst used is very specific for use in acrolein oxidation, the other components aside from steam and air were accounted as inerts.

The complete set of reaction rates considered in the first and second reactors are given in the Appendix A.

### 2.4. Flammability Limits

Operating outside the lower and upper flammability limits (LFL and UFL) is always recommended to avoid any potentially hazardous situation. Nevertheless, this range will tend to change depending on the temperature, pressure and inert gas concentration inside the reactor. For this reason, it is more convenient to define a minimum oxygen concentration (MOC), below which flame propagation is not possible [21]. Indeed, if the concentration of oxygen is maintained below the MOC, it will be possible to prevent fires or explosions regardless of the concentration of the fuel. The MOC value is expressed in units of volume percentage of oxygen in the mixture of fuel and air.

Propylene, which is a flammable gas present in the feed of the first reactor, has a LFL and an UFL of 2.4 and 11 vol% in air, respectively, under standard conditions [22]. At 25 °C, the MOC in air with nitrogen as the inert gas for propylene as a fuel is 11.5 vol%, while it is 14 vol% when the inert gas is carbon dioxide [21]. The MOC for the maximum possible temperature and pressure of the first reactor was calculated, and it was determined that the reactor should be operated below 7.0 vol% of oxygen. As for the second reactor, there was no need to impose a constraint to avoid flammability. If the flammability limit was satisfied in the first reactor, it was expected that it will be satisfied in the second reactor. Furthermore, water was added into the second reactor, which further diluted the reactor concentration.

## 3. Methodology for Solving a Multi-Objective Optimization Problem

In this investigation, the production of acrylic acid was optimized using a multi-objective optimization approach. A methodology to solve large multi-objective optimization problems is proposed in this paper, so that the disadvantage of computational time can be addressed when finding the optimal Pareto front. For this purpose, three-layer artificial neural networks are proposed as a meta-model to directly predict the objective functions.

The proposed methodology comprises six main steps, as illustrated in Figure 3. The optimization problem is first established (Step 1) by defining the set of objective functions to be minimized or maximized, and the set of decision variables (with their respective allowable ranges). Building a representative model to perform the optimization study in a timely manner relies on the data set obtained experimentally or through a comprehensive model of the process (Step 2). To obtain the most suitable set of data, an experimental design is usually adopted to gain maximum information with a minimum number of experiments. This is particularly important when solving the comprehensive model or doing experiments is very time-consuming. In this investigation, uniform design (UD) was adopted as a first approach to determine the process data (Step 3) required to build the meta-model of the process using ANN (Step 4) from the phenomenological model. Once the surrogate model is built, which consists of one ANN for each objective function, it is used to circumscribe the Pareto domain (Step 5) using a genetic algorithm. In this investigation, a large initial population of 5000 individuals was used. Upon analysis of the prediction performance of the ANNs, possible refinement of the optimization problem is considered, such as adjusting the ranges of the decision variables. Initially, the ranges of the decision variables could be larger to ensure a wide range of operation is encompassed. However, when identifying the initial Pareto domain, some ranges of the decision variables could be reduced to ensure more precision for the meta-model. Usually, it would be necessary to return to Step 1, because many data points in the data set used to generate the initial meta-model contains points outside the region of search. Finally, all Pareto-optimal solutions are ranked (Step 6) using the net flow method (NFM) where the preferences of a decision maker are embedded in relative weights and three threshold criteria to assist in the ranking of the Pareto domain. The optimal solution identified based on the ANN and NFM is then corroborated with an actual experiment or via the comprehensive first-principle based model. More details for some of these steps are addressed in the next subsections, where we describe how each step was applied to the optimization of the AA reaction section.

### 3.1. Definition of the Optimization Problem

The multi-objective optimization problem is based on nine objective functions (OF) impacting the economics of the process. These objective functions are presented in Table 1 with their respective nomenclature, equation, and whether they need to be maximized or minimized. The set of decision or input variables consists of eight variables, which are described in Table 2, including their respective lower and upper limits.

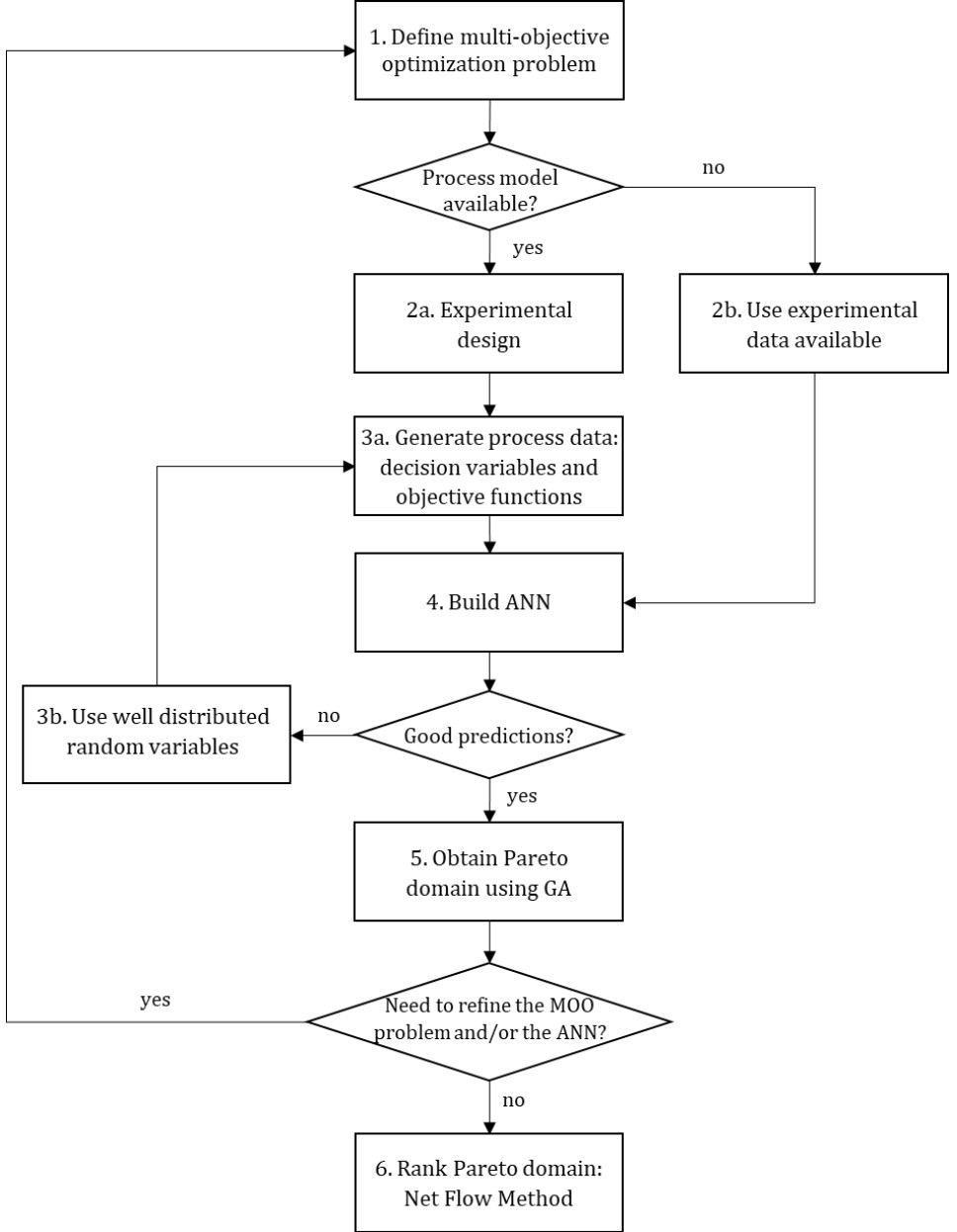

**Figure 3.** Flowchart of the proposed methodology for solving multi-objective optimization using a three-layer artificial neural network (ANN) as meta-model.

**Table 1.** Objective functions for the multi-objective optimization (MOO) of the acrylic acid production process.

| Objective Function | Variable | Max/Min | Equation |
|---|---|---|---|
| Compression Power in C-100 [kW] | $OF_1$ | Min | $\dot{W} = \dfrac{\dot{n}RT_1\left(\left(\frac{P_1}{P_{Feed}}\right)^a - 1\right)}{a \cdot \eta}$ |
| Heat recovery in R-100 [kW] | $OF_2$ | Max | $\dot{H}_{rxn\ j}(T) = \sum\limits_{j} \Delta H_{rnx\ j}(T) \cdot \xi_j$ |
| Productivity in R-100 [kmol/m$^3$h] | $OF_3$ | Max | $Prod = \dfrac{F_{Acrolein}}{V_1}$ |
| Conversion in R-100 [%] | $OF_4$ | Max | $Conv = \dfrac{(F_{reactant\ in} - F_{reactant\ out})}{F_{reactant\ in}}) \times 100$ |
| Compression Power in C-101 [kW] | $OF_5$ | Min | $\dot{W} = \dfrac{\dot{n}RT_1\left(\left(\frac{P_2}{P_1}\right)^a - 1\right)}{a \cdot \eta}$ |
| Heat recovery in R-101 [kW] | $OF_6$ | Max | $\dot{H}_{rxn\ j}(T) = \sum\limits_{j} \Delta H_{rnx\ j}(T) \cdot \xi_j$ |
| Productivity in R-101 [kmol/m$^3$h] | $OF_7$ | Max | $Prod = \dfrac{F_{AcrylicA}}{V_2}$ |
| Conversion in R-101 [%] | $OF_8$ | Max | $Conv = \dfrac{(F_{reactant\ in} - F_{reactant\ out})}{F_{reactant\ in}}) \times 100$ |
| Excess oxygen concentration above LFL | $OF_9$ | Min | $Sum_E = 0$ <br> If $[O_2] > 0.07$ then <br> $Sum_E = Sum_E + ([O_2] - 0.07) * dW$ |

**Table 2.** Decision variables and their allowable ranges.

| Decision Variables | x | Min | Max | References |
|---|---|---|---|---|
| Molar flowrate of propylene [kmol/h] | $F_P$ | 91 | 203 | |
| Molar flowrate of air [kmol/h] | $F_A$ | 433 | 2900 | |
| Molar flowrate of steam [kmol/h] | $F_{S1}$ | 91 | 3047 | |
| Molar flowrate of water vapor [kmol/h] | $F_{S2}$ | 100 | 4000 | |
| Temperature in R-100 [°C] | $T_1$ | 330 | 430 | [17,18] |
| Temperature in R-101 [°C] | $T_2$ | 285 | 315 | [20] |
| Pressure in R-100 [bar] | $P_1$ | 1.05 | 6 | [23–26] |
| Pressure in R-101 [bar] | $P_2$ | 3 | 6 | [23,24,26] |

It is important to notice that the constraint on the molar concentration of oxygen $[O_2]$ in the first reactor, denoted as $Sum_E$, was implemented as a soft constraint expressed as objective function $OF_9$. This objective function represents the minimization of the integration of the excess oxygen concentration above the lower flammability limit. The integration for handling the oxygen concentration as a soft constraint allows for an insight into the set of decision variables that could violate this constraint, and provides more flexibility in the optimization process.

The MOO problem can be expressed mathematically by the following equations:

$$\max_{x}[OF(x)] = \max_{x}[-OF_1(x),\ OF_2(x),\ OF_3(x),\ OF_4(x),\ -OF_5(x),\ OF_6(x),\ OF_7(x),\ OF_8(x),\ -OF_9(x)]$$
$$\text{Subject to } x_{min} \leq x \leq x_{max} \tag{1}$$

### 3.2. Design of Experiments

Process data for the production of AA are required to build a representative model that can be rapidly used to generate the Pareto domain of the process. In this study, a set of neural networks, one for each objective function, were used as a meta-model. Historical process data obtained during the actual operation of a process is one source of information. However, for most processes, historical data usually lack generality, as they only contain information in a very restricted range of operation. This information is usually not sufficient to build a representative model, which is useful for optimization. To obtain process data over a wider range of operation, a common approach resorts to experimental design to adequately cover the range of interest for the operation of the process. An experimental design can be used to obtain process data by conducting experiments on the real process or using a comprehensive first-principle-based model of the process. An important question arises around the number of experimental points necessary and sufficient to properly develop a representative surrogate

model, which in this case means training an ANN to predict each objective function based on a set of design points.

Usually, the design of experiments (DOEs) is used to partition the search domain generated by the independent variables to obtain experimental data that represent the entire domain of interest. The motivation for using DOEs is to determine the underlying relationship that may exist between the decision variables and the objective functions of the process with a restricted amount of data, since a good experimental design should minimize the number of experiments to acquire as much information as possible. Most of the experimental designs assume that the underlying model is known, as it is the case for orthogonal and optimal designs. However, the structure of the model is not always known or may be very complex and highly nonlinear [27]. For such cases, the uniform design (UD) proposed by Fang [28] may be used. UD is a design in which the design points are distributed uniformly on the experimental domain to better capture the relationship between the response and the contributing factors [29].

A large number of UD suggested design points have been tabulated and are available online, where each UD takes the form of $U_n(q^s)$ [30]. In this design, a complete table of the normalized design points can be obtained given the dimensionality of factor space *s* (decision variables), the number of levels *q* of the factors, and the desired number of data points *n*. The normalized information obtained from the selected UD table is then used along with the minimum and maximum values of the decision variables to determine the actual values of decision variables from which the objective functions are calculated. The following uniform designs were used to generate the initial set of process data, divided as learning and validation data sets, to build the ANN: $U_{50}(5^8)$ and $U_{20}(5^8)$, i.e., 50 and 20 design points, which translates to using a ratio of 70/30 for the learning and validation data points.

For more complex cases, where the dimensionality of the input space is large, and a higher number of design points are required in order to develop a good predictive model, one has to resort to another method to define the design points because tables for UD are not available beyond a certain number of design points. In the present investigation, with eight decision variables, it was necessary to use well-distributed random data. This topic will be addressed in more detail in the Results section.

### 3.3. Artificial Neural Networks as Meta-Models

Neural networks are a very versatile meta-model, since they have the plasticity to encapsulate the underlying relationships that exist between input and output process variables based on a number of process data. In this work, the simulated data obtained from the phenomenological process model was used to train a three-layer feedforward artificial neural network (FFANN) for each objective function defined in Section 3.1.

### 3.3.1. ANN Architecture

Figure 4 presents the architecture of the FFANN used in this study, which comprises three layers: the input layer, one hidden layer, and an output layer [12]. The input layer is composed of nine neurons; eight neurons corresponding to the decision variables of the MOO problem, and one bias neuron. The only function of the input neurons is to accept the input variables, normalized between 0 and 1, and fan out these scaled input variables to the processing neurons of the hidden layer. The eight decision variables, as defined in Section 3.1, are the molar flowrates of propylene ($F_P$), air ($F_A$) and steam ($F_{S1}$) to the first reactor, the flowrate of vapor water ($F_{S2}$) to the second reactor, and the operating temperature and pressure of both reactors ($T_1$, $T_2$, $P_1$, and $P_2$).

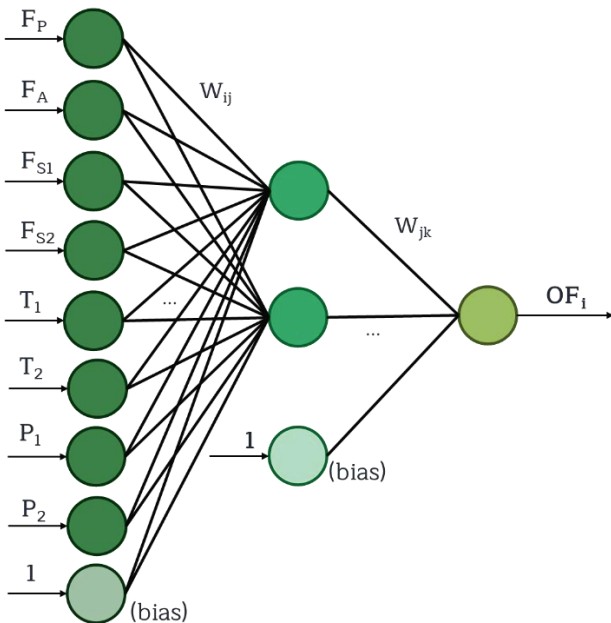

**Figure 4.** Three layer feedforward artificial neural network for the acrylic acid MOO problem.

The hidden layer consists of a number of processing neurons and one bias neuron. The number of hidden neurons is chosen as a compromise to obtain an excellent prediction while avoiding overfitting. Each neuron in the hidden layer, except the bias neuron, performs two simple mathematical operations: (i) the weighted sum of all outputs of the input layer, including the bias neuron; and (ii) a nonlinear transformation of the weighted sum using a sigmoid function [11]. Multiple hidden layers could be included, however, it has been shown that a single hidden layer is generally sufficient for classifying most data sets [31]. Finally, the output neuron performs the same two mathematical functions as the neurons of the hidden layer to predict the output variables that are also scaled between 0 and 1. The prediction of the actual output is obtained by de-normalizing the scaled output. For this case study, ANNs with a single output neuron were used to predict each objective function independently to facilitate the learning process of the networks. This implies that the meta-model thereby consists of nine ANNs in parallel to determine each of the nine optimization criteria defined in Section 3.1.

### 3.3.2. Building the ANN

The experimental data set was divided into learning and validation data to train the networks using a ratio of 70/30 respectively, and 1000 random test points were used to test each ANN after validation. This "second validation" had zero impact on the ANN, as it was only performed to confirm the good adjustment and precision of the neural networks when exposed to new input data. This additional validation was only possible because a phenomenological model of the process was available. To develop an ANN model of the process based on the series of decision variables and objectives, the connection weights were initially assigned small random values and the predicted output was calculated. The sum of squares of the differences between the actual and predicted output was used to change the connection weights in a way to minimize its value. In this investigation, the quasi-Newton optimization method was used to determine the optimal set of connection weights. The sum of squares based on the learning data set was used to adjust the connection weights until the minimum was achieved, which usually requires a good number of iterations. Throughout the learning process, the sum of squares of the errors based on the validation data set was also calculated at each iteration and the selected set of weights was the one associated with this minimum sum of squares. This procedure is an effective way to avoid overfitting and to choose the adequate number of neurons in the hidden layer [32]. Consequently, the $R^2$ value was plotted against the number of hidden neurons

to determine the proper number of hidden neurons for every ANN. The program used to build the networks was coded in FORTRAN.

### 3.3.3. Modified Garson Algorithm

Being a black box model, one would usually expect that no information can be retrieved from the surrogate model besides its predictive ability for a set of input data. Nevertheless, knowing the connection weights of the ANNs, it is possible to determine the percentages of the relative importance of each input variable used to generate a specific output in order to perform a sensitivity analysis. For this purpose, the modified Garson method proposed by Goh [33] was used. The algorithm is given by Equation (1) [33–35]:

$$Q_{ik} = \cfrac{\cfrac{\sum\limits_{j=1}^{L} |W_{ij}W_{jk}|}{\sum\limits_{r=1}^{N} |W_{rj}|}}{\sum\limits_{i=1}^{N} \sum\limits_{j=1}^{L} \left( \cfrac{|W_{ij}W_{jk}|}{\sum\limits_{r=1}^{N} |W_{rj}|} \right)} \tag{2}$$

where $Q_{ik}$ represents the relative influence of the input variable i on the output variable k. $W_{ij}$ is a matrix whose elements are the connection weights between the input neuron i and the hidden neuron j, and $W_{jk}$ corresponds to the connection weights between the hidden neuron j and the output neuron k. i and j are the indices of the neurons in the input and the hidden layers, respectively. The term $\sum\limits_{r=1}^{N} |W_{rj}|$ is the sum of the connection weights between the N input neurons and the hidden neuron j. L stands for the number of hidden neurons connected to the output neuron k. Equation (1) is a modified Garson algorithm, since in order to avoid the counteracting influence due to positive and negatives values of the connection weights, the absolute value of each weight was used [34]. It is important to note that the percentages obtained with this equation are normalized, such that the sum of all of the inputs' relative importance for one objective function adds up to 100.

### 3.4. Optimization Algorithm

To solve the multi-objective optimization problem, gradient-free methods are known to be a good alternative [36]. Evolutionary algorithms that are based on Darwin's theory of survival of the fittest have been widely applied to solve these types of problems [37]. The most well-known and widely used method in this category is the genetic algorithm (GA), developed by Holland in the 1970s [38]. One major drawback of these algorithms is that they require thousands of evaluations of the process model to reach the Pareto front, which consists of only non-dominated solutions. For a process model that is computationally extensive, which is common, it may require days to circumscribe the Pareto domain. In those cases, the use of a meta or surrogate model is very advantageous. In this investigation, ANNs were used as surrogate models for each of the objective functions.

Once trained and validated, the nine ANNs, one for each objective, are used within the MOO algorithm. To solve the optimization problem, the dual population evolutionary algorithm (DPEA), coded in FORTRAN, was used in this work [39,40]. As is the case for other GAs, this algorithm is based on the evolution of a population of individuals, each of which is a solution to the optimization problem [4]. The initial population was comprised of 5000 sets of solutions that were obtained with different sets of decision variables, each decision variable being randomly generated within the permissible ranges. The solutions were then evaluated in pairs to determine the number of times a given solution was dominated. For the next generation, all currently non-dominated solutions were kept along with a fraction of the least dominated solutions. The other solutions were discarded, and the solutions retained from the previous generation were then used to produce new solutions to bring the population to its original number of individuals. This procedure was repeated until

5000 non-dominated solutions were obtained. Using a surrogate model, the MOO converged rapidly, and the 5000 non-dominated solutions lead to a well-defined Pareto domain.

### 3.5. Ranking of the Pareto Domain

The Pareto domain was circumscribed without any bias, that is, with no preferences given to any of the objectives, apart from specifying if a given objective needs to be minimized or maximized. It is obvious that some Pareto-optimal solutions are better than other solutions such that a method is required to rank all Pareto-optimal solutions using some preferences expressed by an expert or decision-maker. In this investigation, the net flow method (NFM) was used. This method requires an expert who has a good knowledge of the process and can give an appreciation on the nature of each criteria. This information is expressed for each objective function via four parameters, namely the relative weight ($W_k$), the indifference threshold ($Q_k$), the preference threshold ($P_k$), and the veto threshold ($V_k$) [41,42]. Using these quantitative parameters, the NFM performs a pairwise comparison of all Pareto-optimal solutions and attributes a score to each one, which then allows all of the solutions to be ranked. An interesting feature of this ranking method is its robustness, which means that changes in the weights will not incur in major changes of the optimal zones [41].

## 4. Results

### 4.1. Construction of the Meta-Model

The initial attempt to develop the nine ANNs, as a surrogate model to represent each of the nine objective functions, was performed with training and validation data sets that were the design points of the uniform design. In this first attempt, 50 and 20 design points were used for the training and validation data sets, respectively. Eight decision variables and five levels were used to separate the ranges of the decision variables, which correspond to $U_{50}(5^8)$ and $U_{20}(5^8)$ for the training and validation data sets, respectively. The coefficients of determination ($R^2$) for each of the nine objective functions (Table 1) are plotted as a function of the number of hidden neurons (Figure 5).

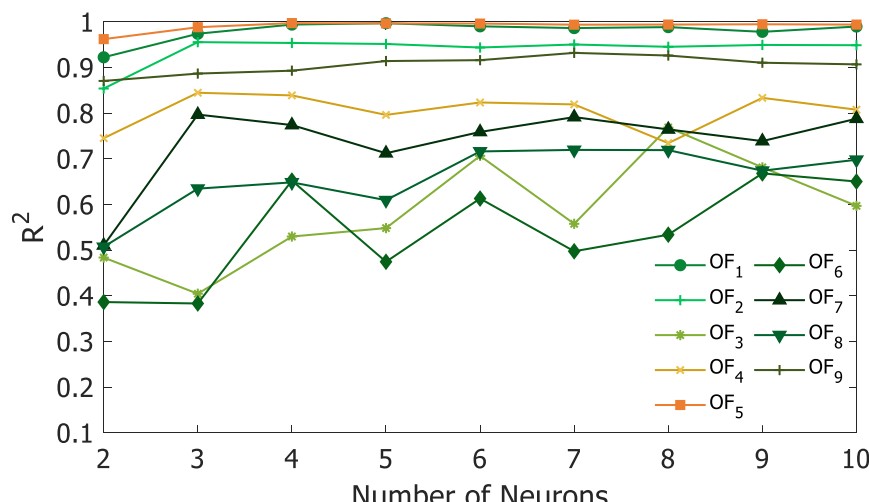

**Figure 5.** $R^2$ values for all the objective functions using 50 learning and 20 validation uniform design (UD) design points vs. Number of neurons in the hidden layer.

Results of Figure 5 show that some objectives, namely the power of the two compressors ($OF_1$ and $OF_5$) and the heat recovery of the first reactor ($OF_2$), are relatively well predicted. However, the ANNs of the other six objectives show poorer predictions with $R^2$ values below 0.90. Furthermore, it is not possible to observe a clear trend for those OFs when one would expect the $R^2$ value to increase as the number of neurons increases. These results suggest that, for these objectives, the number of design

data points is insufficient to allow the ANN to capture the underlying relationships that exist between the decision variables and the objectives. The large number of input variables as inputs to the ANNs also points to the necessity to present the neural networks with richer information. Since the available tables of uniform design are limited to a relatively small number of design points, it was decided to use well-distributed random design points, which offer the possibility of using any desired number of design points.

A series of ANNs were developed for an increasing number of hidden neurons and different numbers of design points. The total number of design points were divided in an approximate ratio of 70:30 for the learning and validation data, respectively. Results for objective functions that showed the best and worst predictions, namely the compression power of the first compressor ($OF_1$) and the heat recovery of the second reactor ($OF_6$) respectively are presented in Figure 6. The total number of design points (training and validation) in the data set varied between 70 and 1430. The predictions for $OF_1$ wwere very good for a relatively low number of hidden neurons and a small number of training data points. Indeed, the very high $R^2$ value indicates that the predictions of the neural network for the compression power of the first compressor were independent of the number of design data points above approximately 140. For the heat recovery of the second reactor ($OF_6$), the coefficient of determination ($R^2$) increased with the number of design points, whereas it was not a function of the number of hidden neurons above five neurons. This trend was more significant for some objectives due to their dependency on the input or decision variables. For example, a simple dependency prevailed for $OF_1$ as it was mainly correlated to the air flowrate and the operating pressure of the first reactor. In contrast, $OF_6$ is a much more complex dependency as it is affected by a larger number of inputs, namely the four input flowrates and the operating temperature and pressure of the first reactor, and thereby requires more data points to capture the underlying relationships between the inputs of the ANN to properly predict this output.

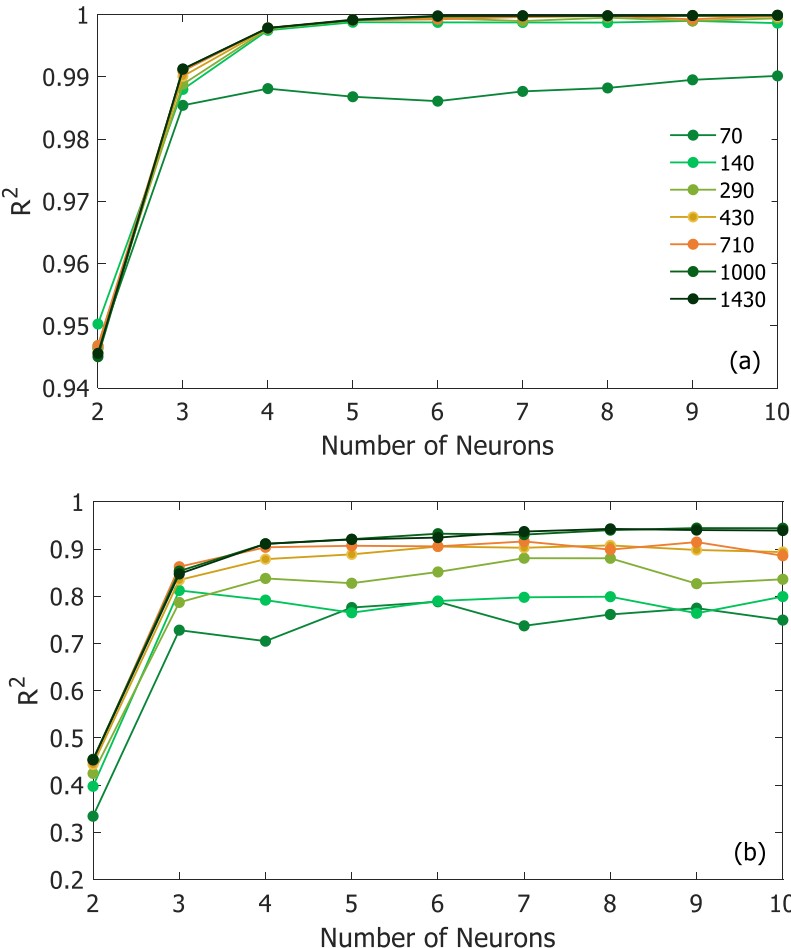

**Figure 6.** $R^2$ values for different number of random data points used for training and validation vs. Number of neurons in the hidden layer for (**a**) compression power in C-100 and (**b**) heat recovery in R-101.

Based on the previous discussion, we performed a sensitivity analysis to extract the contribution of all neural network inputs to explain each output. Numerous techniques have been proposed to provide this information, and they partly alleviate the black box character of neural networks. In this investigation, the modified Garson method was used [35]. Results of this sensitivity analysis are presented in Table 3 in terms of the percentages of the relative importance of the eight decision variables for each of the objective function in the ANNs. It is important to note that the percentages in Table 3 are normalized such that the sum of each row associated to one objective function adds up to 100; if more variables are correlated to an output, the percentages will be obviously lower. First, these results show the paramount importance of the bias neuron of the input layer, which acts in a similar way to the intercept of a linear equation to shift the weighted sum to obtain a better fit. Some strong correlations are logically expected and indicate that the neural networks were trained adequately to capture the underlying behavior of the process. For instance, this is the case for the power of the two compressors that are strongly correlated with the desired pressures and the pertinent flow rates. These sensitivity coefficients offer a valuable introspection on the causal effect of each decision variable on the objective functions.

**Table 3.** Relative importance of the input variables on the objective functions in the selected ANN according to the modified Garson method.

| Objectives/Decision Variables | Relative Importance (%) | | | | | | | | |
| --- | --- | --- | --- | --- | --- | --- | --- | --- | --- |
| | $F_P$ | $F_A$ | $F_{S1}$ | $F_{S2}$ | $T_1$ | $T_2$ | $P_1$ | $P_2$ | Bias |
| $OF_1$ | 1.62 | 17.14 | 2.64 | 1.02 | 1.76 | 1.96 | 41.28 | 1.79 | 30.79 |
| $OF_2$ | 9.07 | 13.75 | 5.23 | 0.79 | 27.76 | 1.93 | 8.22 | 1.33 | 31.90 |
| $OF_3$ | 13.42 | 9.03 | 3.69 | 1.24 | 41.58 | 0.83 | 12.25 | 3.01 | 14.95 |
| $OF_4$ | 5.59 | 18.22 | 7.03 | 4.04 | 36.30 | 1.33 | 19.59 | 2.01 | 5.89 |
| $OF_5$ | 0.62 | 11.45 | 16.24 | 11.69 | 6.36 | 2.22 | 2.02 | 26.39 | 23.02 |
| $OF_6$ | 15.24 | 18.19 | 9.69 | 7.66 | 18.37 | 1.83 | 15.42 | 2.61 | 10.99 |
| $OF_7$ | 17.44 | 34.58 | 5.90 | 6.16 | 10.72 | 1.50 | 8.64 | 2.80 | 12.27 |
| $OF_8$ | 8.64 | 38.13 | 8.07 | 6.60 | 13.50 | 3.42 | 6.16 | 7.38 | 8.10 |
| $OF_9$ | 10.91 | 26.29 | 15.98 | 0.86 | 4.55 | 0.30 | 10.09 | 0.77 | 30.26 |

A large number of ANNs were obtained to represent the nine objective functions. The final selection was made as a compromise of the following criteria: (1) the minimum number of data for training and validating the neural networks; (2) higher than 0.9 values for the coefficient of determination ($R^2$); and (3) the minimum number of neurons. The selected set of nine ANNs, one for each objective function, can now be used as the surrogate model to generate the Pareto domain and find the optimal operating set of decisions variables.

Before proceeding, the quality of the predictions will be examined. Figure 7a,b present the ANN predictions of the conversion of the second reactor ($OF_8$) and the compression power of the first compressor ($OF_1$), respectively, as a function of the values calculated using the phenomenological model. The green points represent the learning data, the orange points correspond to the validation data, and the grey points to the testing data. The testing data were generated using the phenomenological model by randomly selecting the decision variables within their allowable ranges, as defined in Table 2. This data set was used as a "second validation" to confirm the good adjustment and precision of the ANNs when exposed to new input data. As previously mentioned, it had no impact on the meta-model training. The predictions of Figure 7a,b correspond to the ANNs with the lowest and highest $R^2$ values for all data presented: 0.912 and 0.999, respectively. Predictions for the other OFs were very good as well, having $R^2$ values between the two previous values. The predicted conversion in the second reactor (R-101) had the majority of the points near the 45o-line, but with the lowest $R^2$ value due to few scattered points with poor predictions. When examining the sensitivity parameters of Table 3, the two objective functions that are influenced by a larger number of input variables are the conversions of the first and second reactors ($OF_4$ and $OF_8$). As mentioned before, the more an objective function is correlated to a larger number of decision variables, the more learning data points are required to obtain better predictions. As a compromise needs to be made between the $R^2$ value and the number of learning data, a value of $R^2$ above 0.9 was considered a good result.

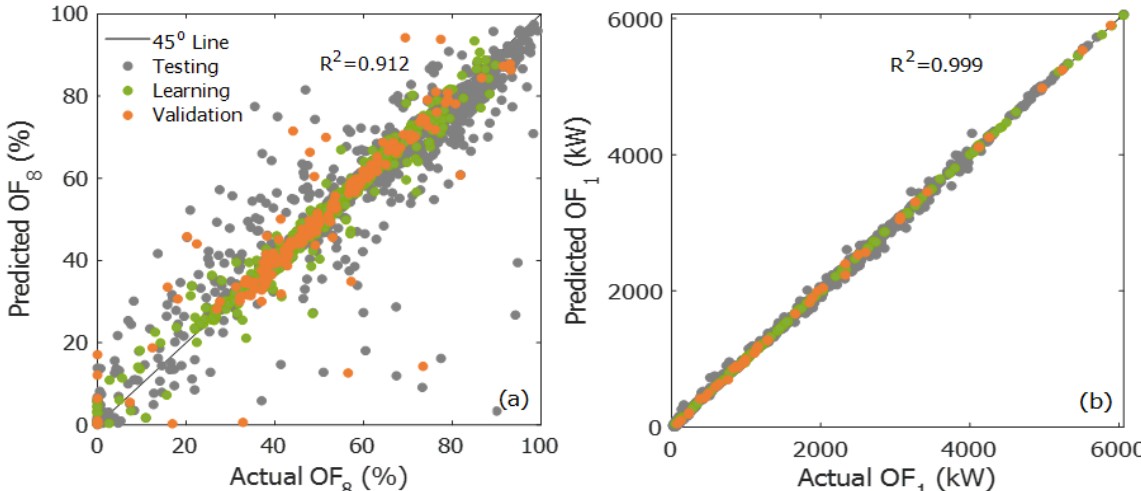

**Figure 7.** Predictions of (**a**) conversion in R-101 and (**b**) compression power in C-100.

*4.2. Multi-Objective Optimization*

After having obtained a good surrogate model, i.e., one consisting of nine AANs, one for each objective function, the Pareto domain was circumscribed with the DPEA and all Pareto-optimal solutions were ranked with the NFM, using both the phenomenological model and the surrogate model for the reactor section to compare the results.

Since there are nine objective functions, the Pareto front is in fact a surface of a nine-dimensional space. To visualize the ranked Pareto domain, it is necessary to resort to two-dimensional projections. In this paper, the ranked Pareto domain of four objective functions are presented for the surrogate and phenomenological models. Figure 8a,b present the Pareto domain projected on the two-dimensional space of the productivity of acrylic acid ($OF_7$) and the heat recovery of the second reactor ($OF_6$), while Figure 8c,d present the projection on the plane of the conversion of propylene ($OF_4$) and heat recovery of the first reactor ($OF_2$). Based on the NFM ranking, the Pareto domain was divided into four different regions: (i) the best solution in red; (ii) Pareto-optimal solutions ranked in the top 5%; (iii) solutions in the next 45%; and (iv) the remaining 50% of the solutions. The best ranked solution of Figure 8b corresponds to a productivity of 1.179 kmol/m$^3$h and a heat recovery of 10,755 kW in the second reactor. When the values of the decision variables, associated with the best-ranked Pareto-optimal solution, were used within the first-principle based model for comparison purposes, the values for $OF_7$ and $OF_6$ were 1.202 kmol/m$^3$h and 11,104 kW, respectively, yielding errors in the vicinity of 2%–3%. This was also the case for the other objective functions, as shown in Table 4. When the phenomenological model is used to circumscribe the Pareto domain and then Pareto-optimal solutions are ranked with NFM, as depicted by Figure 8a, values of 1.2524 kmol/m$^3$h and 11,291 kW were obtained for $OF_7$ and $OF_6$, respectively, for differences of approximately 7% and 5%. The corresponding conversion in R-100 of the best ranked solution was 94.97% and 97.27% for the meta-model and the phenomenological model, respectively, as shown in Figure 8c,d. In contrast, a conversion of 96.25% was predicted when the decision variables of the best-ranked solution identified with the ANNs were used in the first-principle based model.

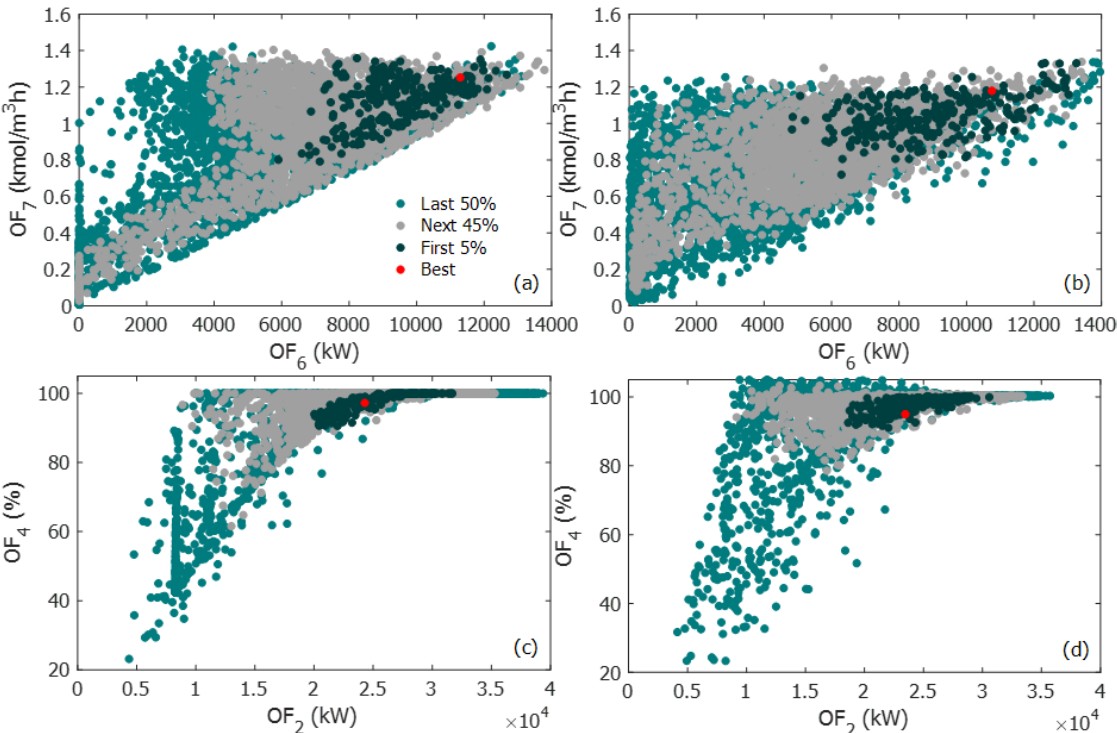

**Figure 8.** Ranked Pareto domain with net flow method (NFM) obtained with: (**a**) and (**c**) the phenomenological model and (**b**) and (**d**) the ANNs.

**Table 4.** Objective functions of the best ranked solution using NFM from the Pareto domain obtained with a population of 5000; $F_P$ = 210.0 kmol/h, $F_A$ = 1507.9 kmol/h, $F_{S1}$ = 206.6 kmol/h, $F_{S2}$ = 100.0 kmol/h, $T_1$ = 697.65 °C, $T_2$ = 580.72 °C, $P_1$ = 1.05 bar and $P_2$ = 4.01 bar.

| Objective Function | Meta-Model | Phenomenological Model | % Difference |
|:---:|:---:|:---:|:---:|
| $OF_1$ | 90.42 | 91.10 | 0.75 |
| $OF_2$ | 23460 | 23700 | 1.02 |
| $OF_3$ | 0.8992 | 0.9866 | 9.27 |
| $OF_4$ | 94.97 | 96.25 | 1.34 |
| $OF_5$ | 6581 | 6794 | 3.19 |
| $OF_6$ | 10755 | 11104 | 3.19 |
| $OF_7$ | 1.179 | 1.202 | 1.93 |
| $OF_8$ | 81.67 | 82.97 | 1.58 |
| $OF_9$ | 0.016 | 0.000 | - |

The Pareto domains generated with the phenomenological and surrogate models were very similar, as illustrated in Figure 8. Occasionally, some minor differences between the two Pareto domains occurred. For instance, there is a small region in Figure 8a that is empty, whereas the same region is covered in Figure 8b. Fortunately, it was observed in this investigation that these discrepancies very often appear in regions where Pareto-optimal solutions are ranked relatively low. These discrepancies were usually due to the inability of the neural network to recognize intrinsic constraints embedded in the code of the first-principle based model and that restricts the operation within those limits. As ANNs are built from experimental data of the model, it will not be possible for the meta-model to explicitly handle the constraints, unless they are treated as soft constraints as was the case for the oxygen concentration. In lieu, it filled the empty region by interpolating the data that is provided to train the ANNs. The best ranked solution as well as the first 5% of the Pareto-optimal solutions were well identified by the meta-model.

The similarity of the Pareto domains (Figure 8) obtained using the phenomenological and surrogate models is a clear indication that the ANNs were able to adequately predict the existing relationship between the decision variables and the objective functions. To make a more complete comparison between the two Pareto domains, the decision variables and the objective functions of the best-ranked solution of the surrogate model were normalized with respect to the best-ranked solution obtained with the phenomenological model, where a value of one was assigned to the latter. The normalized variables are presented in Figure 9a,b. These results clearly show that the use of a surrogate model to perform the MOO is a viable solution, as the great majority of the decision variables and objective functions were very close to the best ranked solution of the phenomenological model. The steam flowrate input was the only variable that has an error above 10%, meaning that using the values obtained from the meta-model will result in 20% more steam usage that what would be required if the first-principle based model was used.

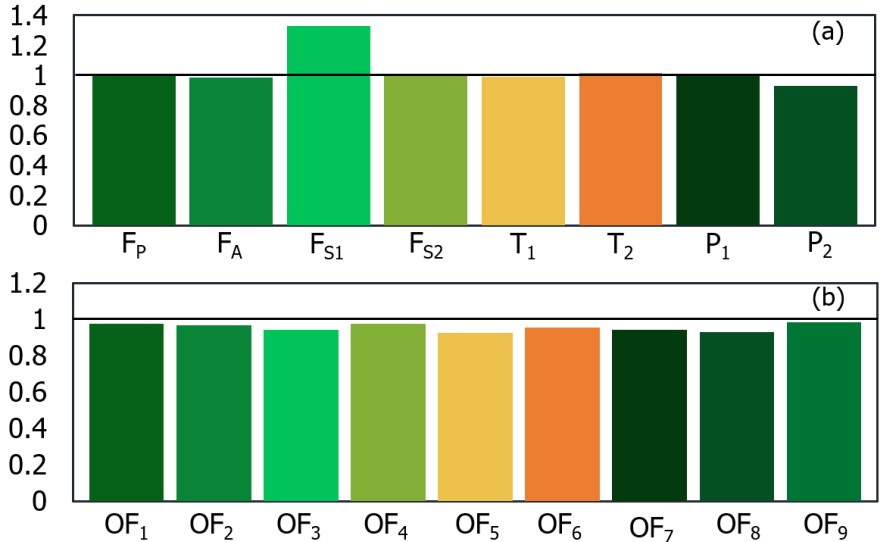

**Figure 9.** Best ranked solution for (**a**) decision variables and (**b**) objective functions using NFM with ANN, normalized with respect to the best solution of the phenomenological model.

In order to compare the resulting solution using the weighted sum method instead, the results of the Pareto domain were used to determine the optimal solution that, as explained earlier, will be a single point in the feasible region for a SOO. All the objectives were assigned a weight of 0.1, except for $OF_3$ and $OF_7$ which were assigned a weight of 0.15. The resulting optimal solution corresponded to a solution ranked in the top 5% when using the NFM method, more specifically the solution ranked 232th out of 5000. The values of the objective functions of this solution are presented in Table 5.

**Table 5.** Objective functions of the solution obtained using the weighted sum method from the Pareto domain; $F_P = 210.0$ kmol/h, $F_A = 1636.5$ kmol/h, $F_{S1} = 495.74$ kmol/h, $F_{S2} = 100.0$ kmol/h, $T_1 = 628.46$ °C, $T_2 = 570.79$ °C, $P_1 = 3.66$ bar, and $P_2 = 6.00$ bar.

| Objective Function | Meta-Model |
|:---:|:---:|
| $OF_1$ | 2170.50 |
| $OF_2$ | 25349 |
| $OF_3$ | 1.1457 |
| $OF_4$ | 100.00 |
| $OF_5$ | 9450 |
| $OF_6$ | 13259 |
| $OF_7$ | 1.331 |
| $OF_8$ | 88.50 |
| $OF_9$ | 0.057 |

The computation time required to circumscribe the Pareto domain via the surrogate model was 38 s. On the other hand, to obtain the Pareto domain optimizing for the reactor section using the first-principle model took 558 s, which means that the optimization process was 15.5 times faster using the ANNs. In this particular instance, one simulation with the first-principle model was relatively fast. In other problems, it may take many days of computation time to obtain a sufficient number of Pareto-optimal solutions, and this is where the methodology proposed in this work would greatly benefit. In addition, once the surrogate model is ready, it also allows a large number of optimization scenarios to be rapidly analyzed. Even if small changes were to be made, the ANN could easily adapt to those changes.

## 5. Conclusions

The aim of this work was to propose an easy-to-follow methodology that would counteract the high computational load of performing multi-objective optimization using first-principle based models. After carrying the optimization process using both the phenomenological model and the meta-model consisting of nine three-layer ANNs in parallel, one for each objective function, the computational time was reduced by a factor of 15.5 while using the meta-model. This approach can also be very useful in terms of hypothesis testing, due to quick convergence of the Pareto domain using ANNs. It should also be noted that the meta-model was able to properly model the existing relationships between the decision variables and the objective functions. This was confirmed by the proper determination of the Pareto domain when comparing the optimization results of the meta-model against the ones obtained with the mathematical model. Results show that the best ranked solution from the NFM using the ANN meta-model was very close to the optimal solution obtained with the phenomenological model. This work therefore successfully demonstrates the advantage of using ANNs as surrogate models to carry out MOO.

**Author Contributions:** Conceptualization, G.C.S. and J.T.; Methodology, G.C.S. and J.T.; Software, G.C.S. and J.T.; Validation, G.C.S., S.O. and J.T.; Formal Analysis, G.C.S.; Investigation, G.C.S.; Writing-Original Draft Preparation, G.C.S.; Editing, G.C.S., S.O. and J.T.; Supervision, S.O. and J.T.; Project Funding Acquisition, J.T. All authors have read and agreed to the published version of the manuscript.

**Funding:** This research was funded by the National Sciences and Engineering Research Council of Canada under the Discovery Grant Program (Grant Number RGPIN 004572).

**Conflicts of Interest:** The authors declare no conflict of interest.

## Nomenclature

| | | |
|---|---|---|
| AA | Acrylic acid | - |
| Ac | Acrolein | - |
| Ace | Acetaldehyde | - |
| AceA | Acetic acid | - |
| ANN | Artificial neural network | - |
| $C_i$ | Molar concentration of i | $kmol/m^3 cat$ |
| CWR | Catalytic wall reactor | - |
| DOE | Design of experiments | - |
| $F_A$ | Molar flowrate of air | kmol/h |
| $F_P$ | Molar flowrate of propylene | kmol/h |
| $F_S$ | Molar flowrate of steam/water vapor | kmol/h |
| LFL | Lower flammability limits | vol% |
| MOC | Minimum oxygen concentration | vol% |
| MOO | Multi-objective optimization | - |
| $\dot{n}$ | olar flowrate | kmol/h |
| NFM | Net flow method | - |
| OF | Objective function | - |
| $n_i$ | Order of reaction j | - |
| $p_i$ | Partial pressure of i | $bar^{nj}$ |
| P | Pressure | bar |
| PBR | Packed bbed reactor | - |
| Prop | Propylene | - |
| R | Gas constant | J/mol K |
| $R^2$ | Coefficient of determination | - |
| T | Temperature | °C |
| UD | Uniform design | - |
| UFL | Upper flammability limits | vol% |
| V | Volume | $m^3$ |
| W | Catalyst weight and connection weight of the ANN | kg |

## Greek Symbols

| | | |
|---|---|---|
| η | Efficiency of the compressor | % |
| ξ | Extent of reaction | kmol/h |

## Subscripts

| | |
|---|---|
| 1 | First reactor |
| 2 | Second reactor |
| Feed | Feed stream to first reactor |
| i | Index of the input layer of the ANN |
| j | Index of the hidden layer of the ANN |
| k | Index of the output layer of the ANN |

## Appendix A  Set of Rate Equations for Each Reaction

The parameters for the reaction rates involved in the first reactor are presented in Table A1 and the rate law equations are below.

**Table A1.** Experimental parameters for the rate law of acrolein formation [18].

| | Constants | Value | Units |
|---|---|---|---|
| $r_1$ | $k_{1Red,o}$ | 0.0628 | $\left[\frac{kmol}{kg \cdot s \cdot bar}\right]$ |
| | $k_{1Ox,o}$ | 16,000 | $\left[\frac{kmol}{kg \cdot s \cdot bar^{0.75}}\right]$ |
| | $\alpha_{H2O}$ | 8.2 | $\left[\frac{1}{bar}\right]$ |
| $r_2$ | $k_{2,o}$ | 2.3200 | $\left[\frac{kmol}{kg \cdot s \cdot bar^{0.86+0.3}}\right]$ |
| $r_3$ | $k_{3,o}$ | 0.0150 | $\left[\frac{kmol}{kg \cdot s \cdot bar}\right]$ |
| $r_4$ | $k_{4,o}$ | 1.4700 | $\left[\frac{kmol}{kg \cdot s \cdot bar^{0.73}}\right]$ |
| $r_5$ | $k_{5,o}$ | 0.0363 | $\left[\frac{kmol}{kg \cdot s \cdot bar}\right]$ |
| | $K_{H2O}$ | 1.9 | $\left[\frac{1}{bar}\right]$ |
| $r_6$ | $k_{6,o}$ | 0.00034 | $\left[\frac{kmol}{kg \cdot s \cdot bar}\right]$ |
| | $K_{H2O}$ | 1.9 | $\left[\frac{1}{bar}\right]$ |
| $r_7$ | $k_{7,o}$ | 1.3800 | $\left[\frac{kmol}{kg \cdot s \cdot bar}\right]$ |
| | $K_{H2O,AA}$ | 55.1 | $\left[\frac{1}{bar}\right]$ |
| $r_8$ | $k_{8,o}$ | 0.00038 | $\left[\frac{kmol}{kg \cdot s \cdot bar}\right]$ |
| $r_9$ | $k_{9,o}$ | $4.75 \times 10^9$ | $\left[\frac{kmol}{kg \cdot s \cdot bar}\right]$ |

The following equations correspond to the rate law of the acrolein formation (Equations (A1) and (A2)). As previously mentioned, the expression for the reaction rate from propylene to acrolein will change depending on whether the temperature is below or above 360 °C. Below 360 °C, if the oxygen concentration increases, the formation of acrolein is accelerated. On the other hand, if the temperature is above 360 °C, the catalytic reduction by propylene will be the rate-determining step.

$$\text{If } T_1 \geq 360\,°C: \; r_1 = k_{1Red,o} \cdot e^{-\left(\frac{39600}{RT}\right)} \cdot p_{Prop} \tag{A1}$$

$$\text{If } T_1 < 360\,°C: r_1 = k_{1Ox,o} \cdot e^{-\left(\frac{114000}{RT}\right)} \cdot p_{O_2}^{0.75} \cdot \left(2 - e^{-\left(\alpha_{H2O} \cdot p_{H2O}\right)}\right) \tag{A2}$$

The side reactions are described using power-law expressions (A3) to (A10). Equations (A6)–(A8) consider water in the adsorption term of the hyperbolic rate expressions since water suppresses the formation of both carbon oxides and leads to higher yields of acrolein and acrylic acid.

$$r_2 = k_{2,o} \cdot e^{-\left(\frac{72500}{RT}\right)} \cdot p_{Ac}^{0.86} p_{O_2}^{0.30} \tag{A3}$$

$$r_3 = k_{3,o} \cdot e^{-\left(\frac{52400}{RT}\right)} \cdot p_{O_2} \tag{A4}$$

$$r_4 = k_{4,o} \cdot e^{-\left(\frac{86700}{RT}\right)} \cdot p_{O_2}^{0.73} \tag{A5}$$

$$r_5 = \frac{k_{5,o} \cdot e^{-\left(\frac{60900}{RT}\right)} \cdot p_{O_2}}{1 + K_{H_2O} \cdot p_{H_2O}} \tag{A6}$$

$$r_6 = \frac{k_{6,o} \cdot e^{-\left(\frac{38100}{RT}\right)} \cdot p_{Prop}}{1 + K_{H_2O} \cdot p_{H_2O}} \tag{A7}$$

$$r_7 = \frac{k_{7,o} \cdot e^{-\left(\frac{82900}{RT}\right)} \cdot p_{O_2}}{1 + K_{H_2O,AA} \cdot p_{H_2O}} \tag{A8}$$

$$r_8 = k_{8,o} \cdot e^{-\left(\frac{14900}{RT}\right)} \cdot p_{Ace} \tag{A9}$$

$$r_9 = k_{9,o} \cdot e^{-\left(\frac{178700}{RT}\right)} \cdot p_{AceA} \tag{A10}$$

As for the parameters of the reaction rates involved in the second reactor, they are presented in Table A2 and the rate law equations are below.

**Table A2.** Experimental parameters for the rate law for acrylic acid formation [20].

|  | Constants | Values | Units |
|---|---|---|---|
| $r_{10}$ | $k_{10,o}$ | 19436 | $\left[\frac{1}{s}\right]$ |
|  | $K_o$ | $9.78 \times 10^{-6}$ | $\left[\frac{1}{s}\right]$ |
| $r_{11}$ | $k_{11,o}$ | 49070 | $\left[\frac{1}{s}\right]$ |
|  | $K_o$ | $9.78 \times 10^{-6}$ | $\left[\frac{1}{s}\right]$ |

The following equations correspond to the rate law of the acrylic acid and $CO_2$ formation:

$$r_{10} = \frac{k_{10,o} \cdot e^{-\left(\frac{55019.6}{RT}\right)} \cdot C_{Ac}}{1 + K_o \cdot e^{-\left(\frac{-31421.84}{RT}\right)} \cdot C_{AA}} \tag{A11}$$

$$r_{11} = \frac{k_{11,o} \cdot e^{-\left(\frac{72006.64}{RT}\right)} \cdot C_{Ac}}{1 + K_o \cdot e^{-\left(\frac{-31421.84}{RT}\right)} \cdot C_{AA}} \tag{A12}$$

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
