# Peer review of "Methodology to Solve the Multi-Objective Optimization of Acrylic Acid Production Using Neural Networks as Meta-Models"

_processes, doi:10.3390/pr8091184_

Round 1
Reviewer 1 Report
I would revisit the Abstract, namely to give it a bit more context to the problem being addressed. I would structure it around the following lines: clearly introduce the research domain/context, describe the problem being addressed, how does it is traditionally handled, and then describe how this work will handle it differently (the novelty of the paper). In other words, use some of the introductory material already present in the Introduction, so that the reader can identify the application domain and associated problem right from reading the Abstract. One more yet minor comment: In the first sentence, I would write "Solving first-principle models can be..." instead of "could be".
I have no expertise in chemistry, thus I focused my review on the meta-modeling and optimization parts.
Figure 3 is quite illustrative of the entire experimental design. What are the main differences between getting good/bad predictions (after step 4) and the need for refining the ANN model (after step 5)? I understand that if you want to refine the ANN, you start the entire process again from the start (back to step 1), but why not do a shortcut to step 3b instead? I am not saying that it is wrong, I am just opening up the discussion.
After step 4, how is the prediction performance assessed? Is it using a pre-defined and fixed test/validation dataset? Perhaps the validation set mentioned in section 3.2, U_20(5^8)? Or maybe I missed this part.
In section 3.3, I understand what the authors mean by "potent model", but wouldn't "powerful" be more correct? This is a minor detail, just a suggestion of mine.
In general, I think the paper is well-written and organized. It has a good presentation too. The presented results seem very interesting.
These are my comments/questions for now.
Reviewer 2 Report
This work presents a neuron network technique to solve a multi-objective optimization problem. The probe process to evaluate this technique is the production of acrylic accid. The produced results were compared with ASPEN results. Despite the fact that there are no real raw experimental data for the evaluation, I believe that this work exhibits novelty and it could be published.
Reviewer 3 Report
The subject and content of the article is interesting, presented in the reliable and comprehensive way.
- Complete the discussion in chapter 4 of the article (comparison with literature data), please.
- Complete the references, please.
Round 2
Reviewer 1 Report
The authors have addressed all my questions/concerns quite well and improved the paper whenever necessary.
Minor comment: Table 2 caught my attention due to the lack of references in the first four entries. Any special reason for this?
I am happy with this version.
Author Response
Table 2 provides the bounds for the decision variables. These bounds are normally chosen such that the decision variables associated to the Pareto domain are enclosed in these bounds. In this study, the minimum required production of acrylic acid was set to 60000 metric tons/yr. Assuming a stream factor of 0.962 which corresponds to 50 weeks of operation per year, the production of acrylic acid in an hour basis was 91.40 kmol/h. If 100% conversion of acrolein is assumed for the second reactor towards acrylic acid, the minimum amount of acrolein required is 91.40 kmol/h. Furthermore, assuming a yield of 90% to acrolein and 100% conversion of propylene in the first reactor, the minimum amount of propylene required would be 101.56 kmol/h.
To set the bounds for the propylene flow, ‐10% and +100 of the minimum value was used to set the lower and the upper bounds. Steam is also present in the feed and the amount will normally be about 1 to 15 moles per mole of propylene [8][9]. As for the oxygen, a molar ratio of 1 to 3 moles per mole of propylene is preferred [8]. Using the steam and the oxygen ratios, the lower bound for both was kept at the lowest amount required and for the upper bound, the highest value of the ratio was used. As for the water vapour that is added to the second reactor, it was given a very wide range to allow for further dilution of the mixture going into the second reactor [10].
The references for the flow rates truly depends on our choice of production. That is why no references were included. The other decision variables do not depend on the rate of production and we decided to provide ranges that will encompass typical values at which a typical system would be operated and we referenced these variables.
Reviewer 3 Report
I accept the Authors' improvement of the manuscript. Thank you.
Author Response
We thank the reviewer for his/her comment.